# Influence of Surface Ligands on Charge-Carrier Trapping and Relaxation in Water-Soluble CdSe@CdS Nanorods

**Mathias Micheel** [1],[†] , **Bei Liu** [1,2,†] and **Maria Wächtler** [1,2,]*

[1] Leibniz Institute of Photonic Technology (IPHT), Department Functional Interfaces, Albert-Einstein-Straße 9, 07745 Jena, Germany; mathias.micheel@leibniz-ipht.de (M.M.); bei.liu@uni-jena.de (B.L.)
[2] Institute of Physical Chemistry and Abbe Center of Photonics, Friedrich Schiller University Jena, Helmholtzweg 4, 07743 Jena, Germany
* Correspondence: maria.waechtler@leibniz-ipht.de
† These authors contributed equally.

**Abstract:** In this study, the impact of the type of ligand at the surface of colloidal CdSe@CdS dot-in-rod nanostructures on the basic exciton relaxation and charge localization processes is closely examined. These systems have been introduced into the field of artificial photosynthesis as potent photosensitizers in assemblies for light driven hydrogen generation. Following photoinduced exciton generation, electrons can be transferred to catalytic reaction centers while holes localize into the CdSe seed, which can prevent charge recombination and lead to the formation of long-lived charge separation in assemblies containing catalytic reaction centers. These processes are in competition with trapping processes of charges at surface defect sites. The density and type of surface defects strongly depend on the type of ligand used. Here we report on a systematic steady-state and time-resolved spectroscopic investigation of the impact of the type of anchoring group (phosphine oxide, thiols, dithiols, amines) and the bulkiness of the ligand (alkyl chains vs. poly(ethylene glycol) (PEG)) to unravel trapping pathways and localization efficiencies. We show that the introduction of the widely used thiol ligands leads to an increase of hole traps at the surface compared to trioctylphosphine oxide (TOPO) capped rods, which prevent hole localization in the CdSe core. On the other hand, steric restrictions, e.g., in dithiolates or with bulky side chains (PEG), decrease the surface coverage, and increase the density of electron trap states, impacting the recombination dynamics at the ns timescale. The amines in poly(ethylene imine) (PEI) on the other hand can saturate and remove surface traps to a wide extent. Implications for catalysis are discussed.

**Keywords:** colloidal nanocrystals; transient-absorption spectroscopy; photoluminescence spectroscopy; charge transfer; charge trapping; surface ligands

## 1. Introduction

Colloidal semiconductor heteronanostructures, e.g., CdSe@CdS dot-in-rod nanorods (NR), have shown great potential as photosensitizers in artificial water splitting, especially for the hydrogen evolution reaction (HER) [1–4]. First, upon light excitation, an electron-hole pair is generated. In quasi-type-II NR, i.e., with a CdSe core diameter < 2.8 nm [5–7], in the lowest energetic exciton state the hole is confined in the CdSe seed, while the electron is delocalized over the entire rod [8]. This localization of the charge carriers supports exciton dissociation and charge transfer to a catalyst bound to the NR surface, e.g., metal or semiconductor nanoparticles or molecular catalysts [9], mediating the reduction of protons to molecular hydrogen. Further, the localization of the hole in the seed, well separated from the electron accepting catalytic reaction center, supports long-lived

charge separation, directly translating into high quantum efficiencies for hydrogen production [4]. For quasi-type-II CdSe@CdS NR functionalized with metal particles at one tip, quantum efficiencies of hydrogen production of several tens of percent to even unity have been reported [1,10–12].

Colloidal nanoparticles such as CdSe@CdS NR cannot be considered as purely inorganic materials, as organic surface ligands mediate the growth of the nanostructure during synthesis and remain an essential part of the final structure, stabilizing them in dispersion. For CdSe@CdS NR, the state-of-the-art method for synthesis is the seeded growth approach [13,14], yielding trioctylphosphine oxide (TOPO) capped NR as final product (in this work, LIGAND-NR will be used in short for LIGAND-capped nanorods while LIGAND will always refer to the free ligand). TOPO-NR are dispersible in non-polar organic solvents such as toluene, chloroform, or hexane, but non-dispersible in polar solvents. However, for application as sensitizers in the light-driven HER, nanorods have to be dispersible in water. For this purpose, the ligand sphere on the nanorods has to be exchanged with hydrophilic ligands making the particles dispersible in water [15].

Hydrophilic ligands can be classified by their anchoring group to the NR surface and their hydrophilic segment [16]. A common choice is mercaptoalkanoic acids (MAA), such as the monothiol 11-mercaptoundecanoic acid (MUA), the dithiol dihydrolipoic acid (DHLA), or the amino acid cysteine [17]. These ligands bind to the nanorod surface via their thiol (or, to be more precise, their deprotonated thiolate) group(s), while at neutral and high pH, the negatively charged carboxylate group stabilizes the dispersed particles in water. MAA-NR lose their colloidal stability at low pH, as the carboxylate group is protonated [15,18]. Thus, systems with this type of ligand are suitable only for photocatalytic systems working at pH > 6 as has been shown for CdSe@CdS nanorods decorated with platinum or nickel particles [12,19–22]. However, some catalysts for the hydrogen evolution reaction such as [FeFe]-hydrogenase-mimics [23–25] or certain metal dichalcogenide nanoparticles, e.g., $MoS_2$, [26,27] require acidic conditions for best performance. Accordingly, for use under these conditions, the hydrophilic segment has to be replaced from a carboxyl group to a moiety allowing for dispersibility at acidic pH or, to realize systems with highest flexibility, dispersibility over a wide pH range. An attractive candidate in this respect is a poly(ethylene glycol) (PEG) chain which has been shown to improve water dispersibility irrespective of pH [28–30] but which has not found wide use in the context of photocatalysis yet (for a PEG-MAA-NR used in the HER, see [19]). A conceptually slightly different approach is the coating of nanorods with water-soluble amphiphilic polymers such as poly(ethylene imine) (PEI), where the imines both serve as anchor group as well as hydrophilic segment. PEI-coating has already been shown to stabilize quantum dots (QDs) [31–36] and NR [19,37] at pH ≤ 7.

Besides mediating the stability in dispersions of various pH, the surface ligands can have severe impact also on electronic properties of the nanostructures, determined by the type of anchoring group. Hence, surface coating can also impact the underlying exciton dynamics in NR. At the surface of the particles, the bonds of the surface forming atoms are not saturated. Intrinsic surface defects such as undercoordinated metal ions, which might act as trap sites for electrons, can be passivated by electron-donating ligands such as TOPO [38]. While thiolate anchor groups act similarly as electron trap passivators, they also introduce trapping sites for holes. In the absence of traps, radiative exciton recombination in the CdSe core occurs efficiently for CdSe@CdS NR. If, however, hole localization to surface trap sites efficiently competes with localization to the CdSe core, a drop in photoluminescence is recorded [38]. Conceptually, this effect could also impact the activity of catalytically active systems, where trapping of charge carriers competes with charge separation and subsequent electron transfer to a catalyst and hole localization to the CdSe core. The effect of surface ligands on photocatalytic efficacy has already been demonstrated: gold-decorated PEI-NR exhibited one (for CdS NR) to two (for quasi-type-II CdSe@CdS NR) orders of magnitude higher photon to hydrogen conversion efficiency than MAA-NR, attributed to enhanced passivation of surface hole traps in PEI-NR [19]. Additionally, in cases where the catalyst domain is not directly in contact with the semiconductor, the choice of the surface ligand can impact and control the distance between the electron accepting

catalyst moiety and the sensitizer. It has been observed for CdS nanorods with surface-proximal hydrogenase that the rate of electron transfer and, accordingly, the hydrogen evolution yield decreases with increasing MAA length [39,40], emphasizing that not only the anchoring group has to be taken into consideration in the design of ligand–NR.

Despite these fundamental observations, only very few studies have tried to unravel the exact influence of different surface coating on the elementary photoinduced exciton dynamics of NR. To close this gap, we present a comparative study on the influence of different commonly employed ligands rendering NR dispersible in water (Figure 1) on the optical properties and exciton dynamics of CdSe@CdS NR. The ligands reported here, MUA, poly(ethylene glycol) 2-mercaptoethyl methyl ether (HS-PEG), DHLA, dihydrolipoic acid poly(ethylene glycol) ester (DHLA-PEG), and PEI can be classified by their anchor groups, namely mono-thiols (MUA, HS-PEG), di-thiols (DHLA, DHLA-PEG), and amines (PEI), as well as their hydrophilic segment, namely carboxylate (MUA, DHLA), PEG chain (HS-PEG, DHLA-PEG), and amines (PEI). We investigated NR with a combination of steady-state and time-resolved absorption and photoluminescence spectroscopy to obtain information on exciton localization and relaxation dynamics in these NR. Results obtained with these surface ligands are compared to the benchmark system TOPO-NR, a phosphine oxide, in toluene. As photocatalysis requires sophisticated and optimized experimental protocols, we opted for photoluminescence measurements as an indicator for the efficiency of charge trapping.

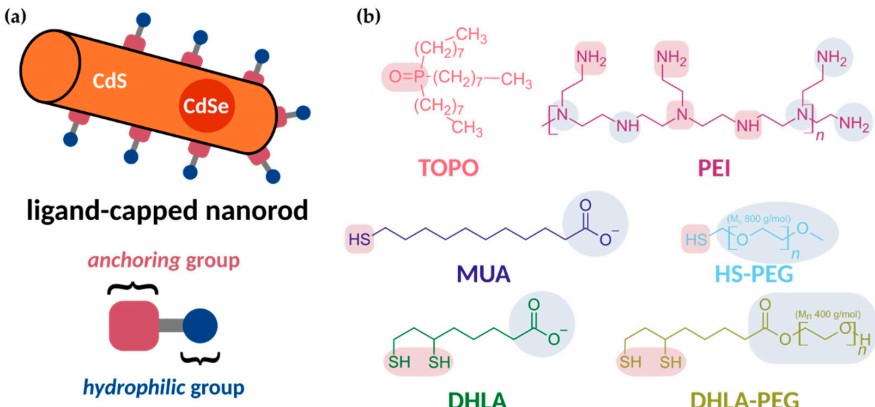

**Figure 1.** (**a**) Schematic sketch of a CdSe@CdS nanorods (NR) with ligands attached on the surface. The red rectangle represents an anchoring group to the surface, the blue one a hydrophilic functionality of the ligand. (**b**) Chemical structures of the ligands used in this work. While NR capped with trioctylphosphine oxide (**TOPO**) are only dispersible in nonpolar organic solvents, 11-mercaptoundecanoic acid (**MUA**), poly(ethylene glycol) 2-mercaptoethyl methyl ether (**HS-PEG**), dihydrolipoic acid (**DHLA**), dihydrolipoic acid poly(ethylene glycol) ester (**DHLA-PEG**), or hyperbranched poly(ethylene imine) (**PEI**) capped NR can be dispersed in water. In all structures, the anchoring groups used in this work, namely monothiols, bithiols, phosphine oxide, and amines, are highlighted in red, while hydrophilic groups, namely carboxylates, poly(ethylene glycol) (PEG) and amines, are highlighted in blue. For PEI, amines can in principle fulfill both roles.

## 2. Results

### 2.1. Particle Synthesis and Ligand Exchange

The particles were synthesized by the seeded growth approach [13] with CdSe quantum dots with a diameter of 2.2 nm as seed, yielding TOPO-NR of a length of circa 30 nm and a diameter of circa 4.5 nm (for the two batches investigated: 30.2 ± 2.3 nm and 4.6 ± 0.5 nm, and 29.3 ± 2.7 nm 4.3 ± 0.5 nm; Figure S1). The original surface ligands were exchanged following protocols adapted from procedures reported in literature. Exchange with MUA [4], HS-PEG, and DHLA (both modified from [4]) was performed under basic conditions, while DHLA-PEG [41] (as basic conditions could

cleave the ligand's ester bond) and PEI [42] were exchanged under neutral conditions at elevated and room temperature, respectively. Infrared (IR) spectroscopy prior and after exchange was used to infer information on the success of the ligand exchange procedure. Further, information on the ligand's binding mode to the nanocrystal surface was derived from the IR spectra (Figures S2–S7 and Tables S1–S6). For the monothiols MUA and HS-PEG as well as the dithiols DHLA and DHLA-PEG, compared to the spectrum of the free ligand the weak S-H stretching vibration at circa 2550 cm$^{-1}$ disappears while the C-S vibration at circa 700 cm$^{-1}$ upon ligand exchange slightly shifts to lower energies, indicating a weakening of the bond, characteristic for binding to the surface via the deprotonated thiolate [43,44] and indicating a successful ligand exchange. PEI contains multiple primary, secondary, and tertiary amines, most of which are not close to the surface. As such, the IR spectrum of PEI remains largely unaffected by ligand exchange with regards to the energetic position of the vibrations. However, vibrations associated with $NH_2$ and C-NH vibrations (3300 cm$^{-1}$) as well as multiple vibrations in the fingerprint region are less pronounced for the bound-ligand compared to the free ligand.

### 2.2. Steady-state UV/Vis Absorption Spectroscopy

First insight on the impact of the ligand sphere on the optical and electronic properties of the NR was gained by steady-state UV/Vis absorption spectroscopy (recorded in toluene for TOPO-NR and water for NR capped with all other ligands). In Figure 2a, spectra of NR with various surface ligands are shown. The absorption spectrum of all NR is characterized by different lowest exciton transition between valence and conduction band levels localized in the CdSe core and different compartments of the CdS shell (Table 1) [45]. A quantitative assignment of different transition was performed on the basis of a five-Gaussian fit of the absorption spectrum (Figures S9–S14 and Table S7). The features at 2.66 and 2.73 eV correspond to the CdS rod based 1Σ ($1\sigma_e$-$1\sigma_h$) and 1Π ($1\pi_e$-$1\pi_h$) exciton transitions, while the broad, low-intensity feature at 2.46 eV is due to the absorption of the CdS shell directly surrounding the CdSe seed (subsequently referred to as "bulb" region) [45,46]. Similarly, CdSe seed absorption can be seen at 2.22 eV. A broad absorption band centered at circa 3 eV is assigned to higher lying transitions, entering a continuum.

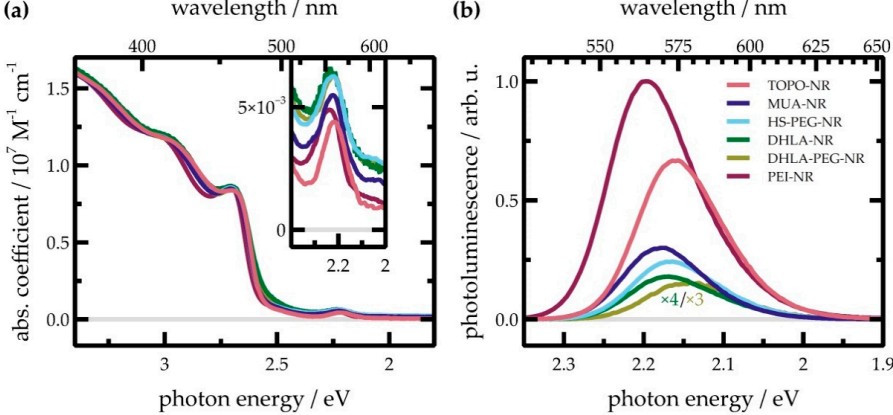

**Figure 2.** Steady-state spectroscopy on NR capped with different ligands. TOPO-capped NR (TOPO-NR) were dispersed in toluene, while MUA-, HS-PEG-, DHLA-, DHLA-PEG-, and PEI-NR were dispersed in water. (**a**) Absorption spectra of investigated samples. Absorption coefficients have been estimated using a procedure based on the spectral shape of the absorption spectra [47]. The inset depicts the region of CdSe core absorption. (**b**) Photoluminescence spectra ($\lambda_{ex}$ = 450 nm) whose integrals have been normalized to their respective photoluminescence quantum yield. Photoluminescence spectra of DHLA- and DHLA-PEG-NR have been enlarged by a factor of 4 and 3, respectively, for better comparison. Area-normalized photoluminescence spectra can be found in the supporting information (Figure S8).

**Table 1.** Steady-state absorption and photoluminescence data of NR with different surface ligands in pure water, TOPO-NR in toluene: $E_{abs,CdS}$ and $E_{abs,CdSe}$ spectral position of the lowest energy excitonic CdS and CdSe seed absorption peak position, $E_{PL}$ spectral position of the peak photoluminescence upon excitation at 3.10 eV (400 nm), and the absolute photoluminescence quantum yields $\Phi_{PL,450}$ and $\Phi_{PL,500}$ upon excitation at 2.76 eV (450 nm) and 2.48 eV (500 nm), respectively. For $\Phi_{PL,500}$, the error is equal to the instrumental error.

| Sample | $E_{abs,CdS}$/eV | $E_{abs,CdSe}$/eV | $E_{PL}$/eV | $\Phi_{PL,450}$ | $\Phi_{PL,500}$ |
|---|---|---|---|---|---|
| TOPO-NR | 2.70 | 2.22 | 2.16 | 0.58 ± 0.09 | 0.77 ± 0.02 |
| MUA-NR | 2.71 | 2.22 | 2.18 | 0.25 ± 0.05 | 0.47 ± 0.02 |
| HS-PEG-NR | 2.71 | 2.23 | 2.17 | 0.21 ± 0.07 | 0.39 ± 0.02 |
| DHLA-NR | 2.70 | 2.23 | 2.17 | 0.04 ± 0.02 | 0.15 ± 0.02 |
| DHLA-PEG-NR | 2.70 | 2.22 | 2.14 | 0.05 ± 0.02 | 0.21 ± 0.02 |
| PEI-NR | 2.71 | 2.24 | 2.19 | 0.91 ± 0.12 | 0.87 ± 0.02 |

Subtle differences between the exact positions of the excitonic peaks between the NR with varying surface ligands can be observed. Notably, all absorption features associated with the first excitonic CdS absorption in PEI-NR are shifted towards higher energies by 10–20 meV compared to TOPO-NR, whereas NR with other ligands show no remarkable shift, indicating no impact on the electronic and structural integrity of the nanostructure. Similarly, the CdSe seed absorption is blue-shifted by 20 meV for PEI-NR as well as DHLA-NR, while other ligands blue-shift the absorption by only 10 meV or less.

*2.3. Photoluminescence Spectroscopy*

Even more pronounced shifts in the peak positions can be observed in the photoluminescence spectra (Figure 2b). For TOPO-NR, excitation at 3.1 eV (400 nm), i.e., the CdS rod, results in photoluminescence peaking at 2.16 eV, which corresponds to emission from the band-edge excitonic state of CdSe. For MUA-NR, HS-PEG-NR, and DHLA-NR, the spectral shape and peak position of photoluminescence remains largely unaffected. PEI-coating, on the other hand, hypsochromically shifts the photoluminescence peak by 30 meV to 2.14 eV, while DHLA-PEG-NR exhibit a bathochromic shift of the photoluminescence peak by circa 30 meV. It is noteworthy that for the two different batches DHLA-PEG-NR investigated in this report, the photoluminescence spectrum differs considerably between the batches. For both batches, however, the photoluminescence spectrum is bathochromically shifted (Figure S13).

To further quantify the changes in their emissive behavior, photoluminescence spectra of the NR were reconstructed using two Gaussians which described the shape of the photoluminescence spectra adequately. Photoluminescence spectra were converted to line-shape spectra to allow a quantitative discussion of excited state populations [48]. Spectra of TOPO-NR, MUA-NR, HS-PEG-NR, and DHLA-NR could be described by two Gaussians centered at 2.17 and 2.13 eV, respectively (Figures S9–S14 and Table S8). For PEI-NR, both Gaussians were indeed blue shifted by circa 30 meV (to 2.20 and 2.15 eV), while DHLA-PEG-NR exhibit a redshift by 20 meV (to 2.16 and 2.10 eV). Notably, Gaussians fits for all samples resulted in the same full-width half maxima (FWHM) of ≈100 and ≈140 meV for both Gaussians. The band edge photoluminescence spectrum of the CdSe QDs used as seeds in the NR synthesis can equally be fitted using two Gaussians with very similar FWHM (centered at 2.47 and 2.42 eV, Figure S15), indicating that this is an intrinsic property of the CdSe core. While it cannot be ruled out that this observation stems from a polydisperse distribution of QD sizes, previous reports on CdS and CdSe QDs also noted that the (steady state or time-resolved) photoluminescence spectrum can be described by two or more Gaussians: the high-energy band is then attributed to photoluminescence from the band edge [49–52], whereas the origin of the broader low-energy emission has been attributed to either "charged excitons", i.e., trions [49,50,53], or shallow trap-state emission [51,52]. While it is outside the scope of the report at hand to unravel the photophysics of the multimodal photoluminescence, it can be stated that ligand exchange does not change the

energetic difference between the states involved. The ratio of the areas under the two curves (defined as the ratio of the high energy to low energy emission curve), on the other hand, revealed a minor trend: while for TOPO-NR, it amounts to circa 1.25, MUA- and PEI-NR exhibited a minor increase (to circa 1.5), while all other samples were in the order of 1.15 or, as was the case for DHLA-PEG-NR, unity. The apparent shift of the photoluminescence maximum can then be explained by the different weights of the two photoluminescent states to the entire photoluminescence spectrum. These results indicate gradual changes of density of hole and/or electron trapping sites depending on the surface functionalization.

Trends were also recorded in the photoluminescence quantum yield upon excitation at 2.76 eV (450 nm) $\Phi_{PL,450}$, which was recorded as absolute yields using an integrating sphere as other authors recommended for scattering NR solutions [54]. For TOPO-NR, a moderately high $\Phi_{PL,450,} = 0.58 \pm 0.09$ was recorded. For both MUA-NR and HS-PEG-NR, a huge drop in the photoluminescence quantum yield to $\Phi_{PL,450,} \approx 0.2$ was observed. PEI coating significantly increased $\Phi_{PL}$ to near unity, while a diametrically opposite behavior, i.e., a nearly complete quenching ($\Phi_{PL,450,} \approx 0.05$) of photoluminescence, was observed for DHLA- and DHLA-PEG-NR. The photoluminescence quantum yield is, generally, defined as the ratio of the emissive rate constant and the sum of non-emissive and emissive rate constants, hence represents the weight of radiative recombination of excitons compared to all deactivation processes. Additionally, excitation at 450 nm mainly generates CdS localized excitons. The hole subsequently localizes into the CdSe core (competing with charge trapping to surface sites) and, finally, radiative exciton recombination in the core occurs (competing with intrinsic non-radiative processes such as Auger recombination) [5,55]. Thus, one can define the observed $\Phi_{PL}$ as

$$\Phi_{PL} = \frac{k_{loc}}{k_{loc} + k_{trap}} \cdot \frac{k_{rad}}{k_{rad} + k_{nonrad}} = \Phi_{loc} \cdot \Phi_{rad} \tag{1}$$

with the rate of hole localization $k_{loc}$, rate of trapping $k_{trap}$, the emissive and non-emissive rate of the CdSe core $k_{rad}$ and $k_{nonrad}$, and the respective hole localization efficiency $\Phi_{loc}$ and intrinsic radiative yield $\Phi_{rad}$ for an exciton localized in the CdSe seed. $\Phi_{PL}$ is accordingly a measure for the efficiency of localization of the exciton to the CdSe core [5] (or, inversely, for trapping of charge carriers) and the efficiency of localization in the CdSe core can be determined if $\Phi_{rad}$ is known. One can get a rough estimate of $\Phi_{rad}$ by measuring the photoluminescence quantum yield upon direct CdSe core excitation. We recorded $\Phi_{PL}$ upon excitation at 2.48 eV (500 nm), the onset of CdS absorption, in order to record a full CdSe photoluminescence spectrum, and which is already close enough to pure CdSe absorption to potentially see an effect [5,54]. For all samples except PEI-NR, an increase in $\Phi_{PL}$ upon excitation at 500 nm was recorded, while for PEI-NR, it remained unchanged (Table 1). Assuming under these conditions $\Phi_{loc} = 1$, then $\Phi_{PL}$ equals $\Phi_{rad}$, and localization efficiency $\Phi_{loc}$ for 450 nm excitation can be estimated (Table S9). It is highest for PEI-NR (circa 1.0) and TOPO-NR (0.75), smaller for MUA- and HS-PEG-NR (both circa 0.5) and smallest for DHLA- and DHLA-PEG-NR (circa 0.25).

Following the assumptions above, $\Phi_{PL,500}$ represents the intrinsic radiative rate $\Phi_{rad}$. This intrinsic yield also depends on the type of surface ligand: it can be seen that non-radiative decay processes gain weight when exchanging TOPO with thiols. This effect is most pronounced for the dithiols. Addition of the bulky PEG-moiety leads to (at least for the monothiols) a further reduction of $\Phi_{PL}$. PEI-NR, on the other hand, show the highest contribution of radiative decay expressed in the highest $\Phi_{PL,500}$. In these results, gradual changes of density of hole and/or electron trapping sites depending on the surface functionalization is also reflected.

The recorded trends can be explained by regarding the bonding modes and surface coverage with the various ligands investigated. Thiolates can be either classified according to Green's covalent bond classification scheme [56–58] as an X-type ligand or, due to their three electron lone pairs, as a σ- and π-donating ligand [38]. On a CdS nanocrystal surface, they have two counteracting effects: First, they passivate undercoordinated $Cd^{2+}$, i.e., with a coordination number below four, located on the surface, which would act as electron traps [59]. While this should result in an increase in $\Phi_{PL}$, the two additional lone pairs act as hole traps on the surface, ultimately diminishing $\Phi_{PL}$ due to the

lower localization efficiency [38]. The decrease in intrinsic radiative yield observed for MUA-NR most likely is dominated by hole trapping. For HS-PEG-NR, a second effect comes into play, which is the bulkiness of the PEG-moiety. This limits the number of surface ligands bound to the surface [60] and leads to an increasing ratio between electron traps due to undercoordinated $Cd^{2+}$ and hole traps induced by the thiolate group binding to the surface. This leads to a further decrease in $\Phi_{PL500}$.

For the dithiolates DHLA and DHLA-PEG, the situation is seemingly more complex. Notably, DHLA-NR and DHLA-PEG-NR differ in their emission spectrum: While DHLA-NR emits similar to MUA- or TOPO-NR, DHLA-PEG-NR's emission is significantly red-shifted. The cause for this can be found in their ligand exchange protocols: while DHLA-NR was synthesized at basic conditions (as MUA-NR and HS-PEG-NR), DHLA-PEG-NR had to be synthesized at neutral conditions at elevated conditions, as basic conditions would cleave the ester bond present in the ligand. Performing DHLA ligand exchange also under neutral conditions at elevated temperatures yielded DHLA-NR which also emitted red-shifted (Figure S16). Possibly, the different surface treatment conditions lead to different surface passivations in the samples. However, this does not significantly affect the emission quantum yield. The lower quantum yields and rod-to-seed localization efficiencies in the dithiolates, in a very rough estimate, could be explained by "double" the amount of hole traps on the surface, due to two HS-binding groups per ligand, which also halves $\Phi_{loc}$ compared to the mono-thiolates investigated. Furthermore, for binding of dithiolates, two neighboring binding sites are necessary [44,61], adding constraints to the coverage of the surface. This leads to an increasing amount of electron trapping sites. These constraints might also be the reason that the surface coverage of the sterically demanding DHLA-PEG, similar to HS-PEG, does not significantly impact $\Phi_{PL}$ compared to DHLA.

PEI-NR behave qualitatively different from the other investigated ligands, as they show not only an enhancement of $\Phi_{PL}$ and $\Phi_{loc}$ to near unity, but also a hypsochromic shift of both absorption and photoluminescence by 20 meV. Notably, this shift also includes the exciton absorption related to the CdSe core. While previous studies on alkylamine-capped CdSe quantum dots assigned these changes to etching of Cd surface atoms, a recent report by Morgan and Kelley proposed that amines do not merely passivate surface atoms, but shifts the energetic levels of both valence and conduction band, making hole traps inaccessible [62]. This results in the blue-shift of absorption and emission and a near-unity photoluminescence.

If varying trap state densities interfering with radiative recombination of band-edge states are indeed the source of the observed effects, this should also be reflected in the photoluminescence decay kinetics. The temporal emissive behavior of the NR was probed in more detail by time-resolved photoluminescence spectroscopy (Figure 3). The photoluminescence decay, describing the radiative exciton recombination kinetics, of TOPO-, MUA-, and PEI-NR, could be well described by a monoexponential decay with a single time constant in the order of 20 ns. Rate constants of radiative recombination have been calculated on the basis of Equation (1). For TOPO- and PEI-NR, $k_{rad}$ is in the order of $4 \times 10^{11}$ s$^{-1}$, implying only minor changes in the electronic structure of the samples. On the other hand, it decreases to $1.9 \times 10^{11}$ s$^{-1}$ for MUA-NR. It has been noted that localized trap states lead to reduced radiative rates in CdSe@CdS QDs [63]. As has been discussed above, MUA-capping leads to hole-trap surface states, which then reduces the radiative rate of recombination.

For HS-PEG-NR, the monoexponential decay changes to a biexponential model with a sub-ns and an 18 ns lifetime. The changed photoluminescence decay kinetics compared to MUA-NR can stem from incomplete surface coverage of ligands: Potentially, the PEG chains prohibit full surface coverage due to steric inter-ligand hindrance, leading to undercoordinated Cd atoms on the surface. This introduces surface electron traps, ultimately changing the exciton recombination kinetics. For sub-2-nm CdSe QDs, it has been shown that less than 4 PEG molecules per QD adsorbed to the surface [60] (it has to be noted that the ligand exchange protocol differs from the one used in the report at hand). The longer lifetime, on the other hand, represents the exciton recombination dynamics similar to MUA-NR (with a similar $k_{rad} = 2.2 \times 10^{11}$ s$^{-1}$).

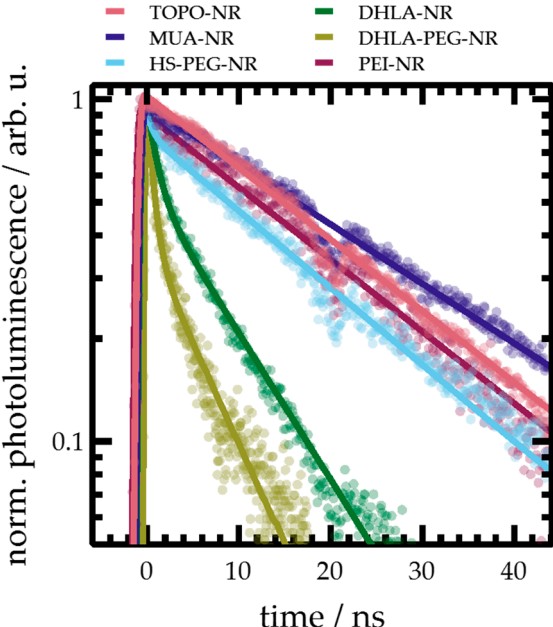

**Figure 3.** Time-resolved photoluminescence decay of samples investigated measured in toluene (TOPO-NR) or water (all other samples). The solid line represents a mono- or biexponential fit of the data (dots), respective fit parameters are summarized in Table 2.

**Table 2.** Time-resolved photoluminescence spectroscopy on NR with different surface ligands. Time constants were obtained via a mono- or biexponential fit. For the biexponential fits, the amplitudes of the respective component are indicated. Additionally, the rate of radiative recombination $k_{rad}$, which was calculated from Equation (1) using $\tau_2$, is also given.

| Sample | $\tau_1$/ns | $A_1$ | $\tau_2$/ns | $A_2$ | $k_{rad}/10^{11}$ s$^{-1}$ |
|---|---|---|---|---|---|
| TOPO-NR | - | - | 19.7 ± 0.4 | - | 3.9 ± 0.2 |
| MUA-NR | - | - | 25.0 ± 0.3 | - | 1.9 ± 0.1 |
| HS-PEG-NR | 0.9 ± 0.2 | 0.31 ± 0.14 | 18.0 ± 2.0 | 0.69 ± 0.14 | 2.2 ± 0.4 |
| DHLA-NR | 0.8 ± 0.3 | 0.43 ± 0.03 | 8.4 ± 2.5 | 0.57 ± 0.03 | 1.8 ± 0.8 |
| DHLA-PEG-NR | 0.6 ± 0.2 | 0.64 ± 0.05 | 7.5 ± 1.1 | 0.36 ± 0.05 | 2.8 ± 0.7 |
| PEI-NR | - | - | 20.1 ± 0.4 | - | 4.3 ± 0.2 |

DHLA- and DHLA-PEG-NR also followed a biexponential decay, which compared to MUA is accelerated with a sub-nanosecond and a 7 ns component. It is noteworthy that both DHLA- and DHLA-PEG-NR, as well as the two different batches of DHLA-PEG-NR, decay near identically, despite their different steady-state photoluminescence spectra. Similar to HS-PEG-NR, the biexponential decay can be explained on the basis of incomplete surface coverage inducing electron trapping sites. It is widely assumed that using DHLA as an anchoring group leads to reduced surface coverage compared to structurally similar MAA (circa 25 % for 4.5 nm CdS QD [44]) and DHLA could either bind to two separate Cd surface atoms or act as a chelating ligand with both thiolates binding to the same Cd atom [44,61]. It is feasible that exchange with DHLA leads only to a partial coverage of the surface, nevertheless with the additional thiolate (compared to e.g., MUA) acting as an additional hole trap decreasing $\tau_2$. In fact, it has been shown that treatment of ZnSe@CdS QDs with $H_2S$ changes the photoluminescence decay from a mono- to a biexponential decay (in addition to pronounced photoluminescence quenching) [64], with adsorbed $S^{2-}$, $SH^-$, or $H_2S$ as hole traps. DHLA could induce similar changes. For both samples, $k_{rad}$ is nearly unchanged from the monothiolate ligands, indicating an increase of non-radiative deactivation pathways. Taking these effects together it is feasible to assume these are related to the increased amount of hole and electron traps on the surface [60].

### 2.4. Transient Absorption Spectroscopy

The data collected so far indicates that ligand-induced surface traps mediate surface-located trap states, which influence the radiative exciton recombination dynamics on a nanosecond-timescale. However, it only partially sheds light on the different localization efficiencies between samples. As charge localization occurs on a sub-nanoseconds timescale [46], transient absorption spectroscopy was employed to deliver information on the temporal evolution of excitons generated by excitation of electrons to the CdS and CdSe conduction band levels [65], helping to identify ligand-induced changes on the exciton relaxation and localization and trapping processes. The excitation wavelength was set to 390 nm, directly exciting mainly the CdS and to a lesser extent the CdSe domain (with a power density small enough to generate one exciton per NR or less) and the sample was probed with a white light continuum spanning from 3.54–2.33 eV (350 nm to 700 nm) up to a delay time of 1700 ps.

Qualitatively, the transient spectra for all NR evolve similarly (Figure 4a shows transient spectra recorded for TOPO-NR at different delay times, representations for all NR see Figures S17–S22). Transient spectra obtained few hundreds of femtoseconds after excitation are characterized by a sharp negative feature peaking at 2.68 eV (by comparison with steady-state absorption spectra assigned to a bleach of the CdS exciton transition), a broad bleach feature centered at 2.21 eV (assigned to the bleach of the CdSe exciton transition) and a slightly positive feature at 2.57 eV, due to the presence of hot excitons generated in the nanorod upon excitation into higher excitonic transitions at 390 nm [45,66]. Within the first picosecond, the amplitudes of both bleach features grow, i.e., become more negative, the positive feature at 2.57 eV vanishes and becomes negative, and a positive excited state absorption feature rises at 2.90 eV instead, which is induced by interaction of the S(e)1S3/2(h) exciton with other exciton bands, inducing a red-shift of these transitions [66,67]. Additionally, a broad, minute positive amplitude was recorded for probe photon energies lower than 2.1 eV (600–750 nm, Figure S23), which builds up within the first ps. Previous studies attributed this spectral feature to the dissociation of excitons into electrons and holes with subsequent trapping of holes on the surface [46,68,69]. The subsequent behavior is characterized by the decay of the CdS bleach feature, accompanied by a small red-shift of the spectral position of the minimum of circa 50 meV, and the decay of the CdSe bleach feature, accompanied by a small blue-shift of the minimum of circa 30 meV. Both bleach features do not fully decay until the end of the temporal detection window employed (limited by the mechanical delay stage). Transient spectra recorded for the different NR are comparable with respect to the spectral evolution and position of bleach minima, with the exception of PEI-NR, which exhibits a blue shift of both CdS and CdSe bleach of circa 30 meV, in accordance with the shift observed in the steady-state absorption spectrum.

A global fit of the transient absorption data delivers a quantitative basis for the subsequent discussion. Since it is known that the features associated with CdS and CdSe decay with different characteristic time constants, the energy ranges of 2.48–2.95 eV (corresponding to a wavelength range of 420–500 nm) and 1.97–2.34 eV (530–630 nm) were fitted separately. For all NR, four time constants and an offset were needed to describe the excited CdS decay (420–500 nm) while the number of components varied for global fitting of the spectral region of CdSe bleach, yielding three time constants for DHLA- and DHLA-PEG-NR and four for the remaining samples (Table 3). In the following, the decay associated spectra $DAS(\tau_n)$, indicating the spectral changes associated with $\tau_n$, and the respective processes will be discussed.

$DAS(\tau_{1,CdS})$ is characterized by a strong positive band at 2.70 eV for all samples, describing the fast sub-picosecond (limited by the temporal resolution of the measurement setup) build-up of the CdS bleach signal. Additionally, a negative amplitude, indicating a rise of the positive signal with the associated time constant, was found at 2.93 eV, the edge of the spectral window used in the fitting procedure. These changes represent the relaxation of "hot" excitons formed upon photoexcitation to the band edge on a sub-picosecond time scale, populating the 1σ level [5,45,66]. This is visible as a build-up of both the CdS and also the CdSe bleach features. The respective $DAS(\tau_{1,CdSe})$ is characterized by a positive amplitude centered at circa 2.2 eV, indicating bleach formation. The CdSe bleach build-up occurs more slowly than the CdS one ($\tau_{1,CdS} \leq \tau_{1,CdSe}$). This is because of an additional

process contributing to the formation of the bleach signal, i.e., the localization of excitons from the CdS shell into the CdSe core due to the quasi-type-II band alignment of the NR investigated, which routinely occurs on a sub-ps to circa 10 ps timescale [10,45,70,71]. Due to similar spectral changes associated with the processes, this process was not discriminated from the hot exciton relaxation process in the fit of the CdSe band, but is represented in $DAS(\tau_{2,CdS})$. This process occurring with time constants $\tau_{2,CdS}$ in the range of 1–3 ps for all samples exhibits a sigmoidal shape with a slightly positive amplitude at 2.64 eV and a broad, negative amplitude at circa 2.77–2.80 eV: These changes describe the bleach recovery associated with the relaxation of an exciton generated in the CdS rod to form excitons localized in the CdSe seed driven by the localization of the generated hole in the CdSe seed [45,67,72], while the formation of a negative amplitude at 2.64 eV is associated with the localization of electrons close to the CdS bulb region [46], for which the bandgap is slightly reduced [72].

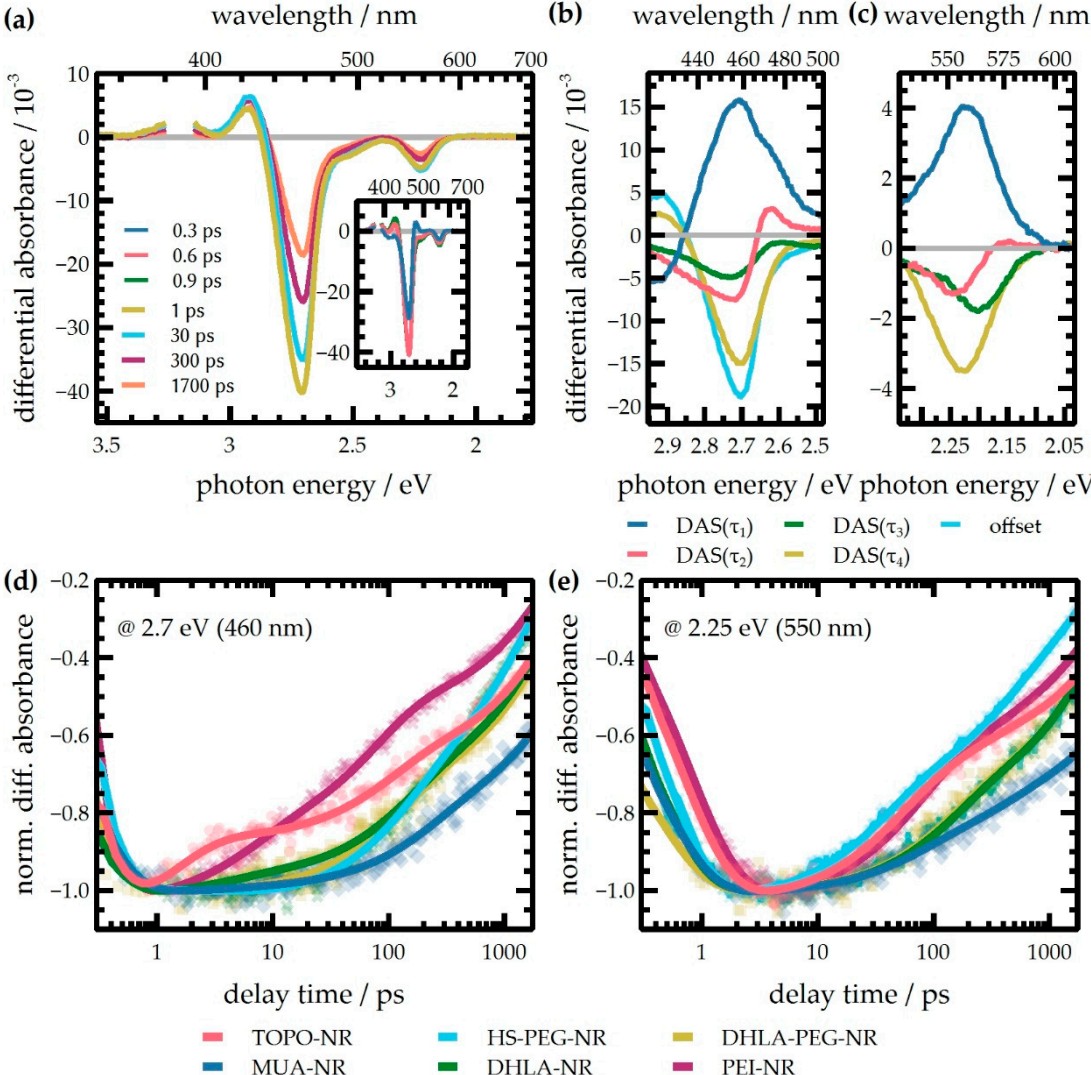

**Figure 4.** Transient absorption spectroscopy on the investigated structures recorded upon 390 nm excitation. (**a**) Transient spectra of TOPO-NR recorded at the indicated delay times. (**b**) and (**c**) Decay Associated spectra obtained via a global fit for TOPO-NR. The spectral regions 2.48–2.95 eV and 1.97–2.34 eV were fitted separately from each other. The data for all other ligands is displayed in the SI. (**d**) and (**e**) Kinetic traces at 2.70 eV (460 nm) and 2.25 eV (570 nm), respectively, recorded for the different samples. Kinetic traces were normalized to −1 at the minimum of the fit function (solid line), while recorded data points are shown as semi-transparent points.

**Table 3.** Transient absorption spectroscopy on NR with different surface ligands. Time constants were obtained via a global fit within the indicated wavelength range. For the range 420–500 nm (indicating changes in excited CdS), four time constants and an offset were needed for all samples. For the range 530–630 nm (indicating changes in excited CdSe), two time constants and an offset were needed for DHLA- and DHLA-PEG-NR, and three and an offset for TOPO-NR, MUA-NR, and HS-PEG-NR, and PEI-NR.

| Ligand | Energy Range | $\tau_1$/ps | $\tau_2$/ps | $\tau_3$/ps | $\tau_4$/ps |
|--------|--------------|-------------|-------------|-------------|-------------|
| TOPO | 2.48–2.95 eV | 0.2 ± 0.0 | 1.0 ± 0.2 | 34 ± 8 | 530 ± 70 |
| | 1.97–2.34 eV | 0.7 ± 0.0 | 47 ± 12 | 190 ± 50 | 6200 ** |
| MUA | 2.48–2.95 eV | 0.2 ± 0.0 | 2.5 ± 0.4 | 78 ± 6 | 650 ± 40 |
| | 1.97–2.34 eV | 0.5 ± 0.1 | 58 ± 15 | 540 ± 170 | 12200 ** |
| HS-PEG | 2.48–2.95 eV | 0.2 ± 0.1 | 3.0 ± 0.6 | 68 ± 4 | 780 ± 30 |
| | 1.97–2.34 eV | 0.5 ± 0.1 | 42 ± 7 | 450 ± 140 | 4000 ** |
| DHLA | 2.48–2.95 eV | 0.2 ± 0.0 | 4.8 ± 1.0 | 86 ± 7 | 820 ± 40 |
| | 1.97–2.34 eV | 0.5 ± 0.1 | — * | 150 ± 20 | 4000** |
| DHLA-PEG | 2.48–2.95 eV | 0.2 ± 0.0 | 1.7 ± 0.2 | 60 ± 10 | 610 ± 160 |
| | 1.97–2.34 eV | 0.4 ± 0.0 | — * | 190 ± 50 | 2500 ** |
| PEI | 2.48–2.95 eV | 0.2 ± 0.0 | 1.3 ± 0.3 | 25 ± 1 | 470 ± 70 |
| | 1.97–2.34 eV | 1.0 ± 0.2 | 74 ± 11 | 270 ± 30 | 4100 ** |

* For DHLA- and DHLA-PEG, only three time constants were needed to describe the temporal evolution of the signal in this spectral region. The assignment to a specific time constant is based on the spectral similarity of the decay associated spectra (DAS) of the different systems investigated. ** The relative error associated with this time-constant is in the order of 20–50%, but was included nonetheless in the fitting procedure.

Parallel to the initial CdS and CdSe bleach formation processes formation of a broad positive amplitude for probe photon energies lower than 2.1 eV (Figure S23), indicative of surface trapping of holes [46,68,69], is observed. This process exhibits a minor ligand-dependence: for all thiol-based ligands, the risetime is in the order of 0.3 ps and below, while TOPO- and PEI-NR exhibit slightly higher risetimes (0.8 and 0.5 ps, respectively). In general, electron localization is driven by prior hole localization: holes trapped to surface sites will prohibit hole-localization-driven electron localization in the seed and effectively keep electrons localized in the rod. Both processes, driven by Coulombic interactions, occur in parallel. Accordingly, the time constant $\tau_{2,\mathrm{CdS}}$ observed for rod to seed localization should be independent of the presence of surface hole traps, in good agreement with the findings described above. For the thiolate-based systems, this process is significantly slowed down compared to TOPO-NR, possibly related to the different solvents (water and toluene, respectively) used: the more polar the solvent, the slower charge separation and localization occur in NR [73]. For PEI-NR, this process occurs on a similar timescale as for TOPO-NR, probably related to the different electronic structure induced by the ligand, negating the solvent effect.

Nevertheless, the efficiency of electron localization to the seed region is directly affected by the presence of hole traps at surface sites, expressed in the efficiency of hole localization $\Phi_{\mathrm{loc}}$. The trend derived from the photoluminescence quantum yields can also be found in the results from transient absorption spectroscopy. A quantitative measure for hole localization in the seed is the relative contribution of the localization component (represented by the relative integrated amplitude of $DAS(\tau_{2,\mathrm{CdS}})$) compared with the remaining three $DAS$ describing CdS rod decay (a conceptually similar approach can be found in ref. [46]). For TOPO- and PEI-NR, this value is in the order of 0.25, whereas it amounts to ~0.10 for all other systems (Figure S24). Notably, these values differ from the localization efficiencies derived from the photoluminescence quantum yield measurements. It has to be emphasized that $\Phi_{\mathrm{loc}}$ is not a sole measure of exciton localization, but is affected by hole and electron trapping, whereas the value derived from the transient absorption measurement is only sensitive to electrons which localize strongly to the CdSe core (which is also affected by sample homogeneity [45,46]) and only an indirect measure of hole localization (or trapping). Still, the trend of lower localization in the thiolate-based ligand system remains visible also with this approach.

After the initial 10 ps, decay of signal amplitudes can be observed for the entire spectral window. Residual decay of CdS rod bleach at delay times > 10 ps, could be well described by two time constants and an offset spectrum. Corresponding *DAS* are spectrally very similar with negative amplitudes at 2.71 and 2.70 eV for all NR except PEI-NR, for which these features are blue-shifted by circa 30 meV in agreement with the positions in the ground state absorption spectra. These decays describe intrinsic decay processes such as recombination processes between holes and electrons involving holes trapped on the surface or the CdS rod [46,68,74,75]. Further, processes involving trapping of electrons at the surface opening additional decay pathways can contribute to this signal decay, if respective trap sites are available in the structures [76]. The kinetics of CdS bleach recovery are not pronouncedly different between samples with the exception of PEI-NR, where it occurs faster. It has been noted that in CdSe@CdS NR, rod-to-seed localization of electrons completes within 100 ps after photoexcitation [46]. As the hole localization efficiency $\Phi_{loc}$ is near unity for PEI-NR, the reason for the fast rod bleach recovery might lie in the efficient charge carrier localization to the seed.

Similarly complex decay dynamics are observed for the CdSe bleach recovery of the initially relaxed exciton which can be described by two (DHLA- and DHLA-PEG-NR) or three (all other samples) decay times. Again, the *DAS* associated with the time-constants are very similar to each other. Knowles et al. have described complex radiative and non-radiative recombination channels in pure CdSe QDs with time constants very similar to those found in our system [76]. CdSe bleach recovery is then associated with competing processes of radiative recombination with hole trapping (for sub-100 ps processes) or electron trapping (for slower processes). The absence of the $\tau_{2,CdSe}$-process in the analysis of the DHLA-based systems could be then rationalized that very efficient and quantitative hole trapping on the sub-100 ps timescale efficiently inhibits radiative recombination, which would contribute to the CdSe bleach recovery. A $DAS(\tau_{2,CdSe})$ for these systems would then only contribute to the bleach recovery with a minute amplitude, which the global analysis of the data might not catch. CdSe bleach recovery continues on a timescale of nanoseconds, due to the radiative and non-radiative recombination of electrons and holes, which is reflected in the time-resolved photoluminescence measurements (vide supra).

## 3. Discussion

In this report, ligands with commonly employed anchor groups, namely mono-thiols, di-thiols, and imines, and hydrophilic segments with different steric demands (carboxylates and PEG) were investigated with regards to their impact on the photophysics of the NR. Particular emphasis was laid on trapping of charge carriers generated upon photoexcitation. Main results are summarized in Figure 5.

Surface ligands can impede charge carrier localization to the CdSe core. Photoinduced rod-to-seed localization of charge carriers is a fast process in NR, occurring within several picoseconds, and efficient rod-to-seed localization has been found to be a key parameter determining catalytic efficiency in systems building on seeded NR [4]. From excitation wavelength dependent photoluminescence quantum yield measurements, we could infer that trapping of holes on the surface efficiently competes with hole localization in mono- and dithiolate-NR, with trapping efficiencies of circa 0.5 and 0.75, respectively. PEI-coating, on the other hand, passivates the surface and changes the electronic structure of the NR that hole-traps are not accessible anymore, giving access to a rod-to-seed localization efficiency of nearly unity. Comparison of the relative amplitudes of the hole-localization driven electron rod-to-seed localization in transient absorption measurements were in good agreement with this finding. Hole localization in the seed in metal tipped seeded NR was shown to improve excited state lifetime [77], which correlates with improved photon-to-hydrogen conversion efficiencies [4]. In line with this, thiolate-NR showed significantly lower catalytic efficiencies compared to, for instance, PEI-NR [19].

Disorder in the ligand shell can be the source of additional electron trap states. Generally, the MAAs investigated behaved nearly identical compared to their PEG-analogues in terms of

photoluminescence quantum yield and charge localization efficiency. However, some distinct changes in the photoluminescence decay, which is an indicator for the radiative recombination of the core-localized exciton on the nanosecond timescale, were recorded. While MUA-NR exhibited a monoexponential decay, it changed to a biexponential emission decay for HS-PEG-NR. This finding was rationalized on the basis that bulky PEG-chains inhibit high surface coverage of the NR with ligands, leaving surface Cd atoms undercoordinated, which then serve as electron trapping sites. Similarly, the sterically demanding DHLA binding group also introduces electron traps, changing the emission decay to a biexponential process, as well as additional hole traps, which reduce the lifetime of the exciton. The intrinsic rate of radiative recombination is, however, the same between all thiolate samples, but reduced compared to TOPO- and PEI-NR.

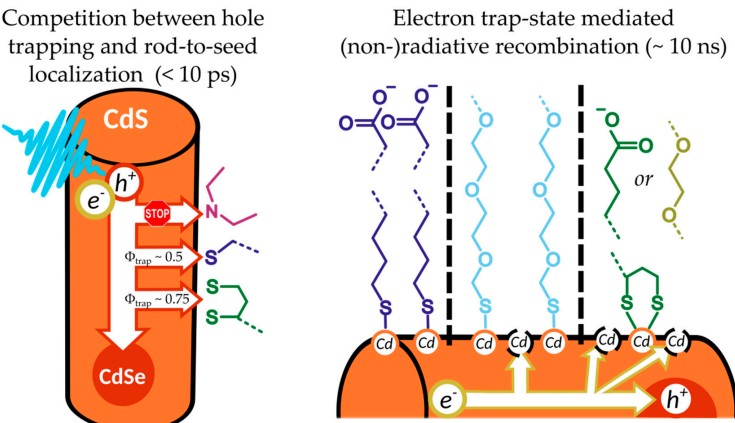

**Figure 5.** Summary of the effect of different water-soluble ligands on the excited state properties of CdSe@CdS NR. On a sub-10-ps timescale, mono- and ditholates offer hole trapping sites, opening a channel for processes competing with exciton localization to the CdSe core, while PEI prevents any trapping. On a ns-timescale, radiative exciton recombination is impaired by non-radiative processes induced by electron trapping sites. These result from the presence of either the bulky hydrophilic chain (PEG) or the anchoring group (in particular, DHLA). PEI (not shown) allows for a near-unity charge recombination, saturating surface defects very efficiently without inducing additional hole trapping sites.

Low surface coverage could be battled by careful choice of the parameters (notably temperature, solvent, or concentration) during the ligand exchange procedure. A recent report on CdSe@CdS QDs demonstrated how careful calibration of a ligand–composite system (namely oleic acid and an alkyl phosphine) can be used to induce order in the ligand sphere, increasing photoluminescence [78]. Similar approaches might be applicable to increase order in the ligand sphere of PEG-ligand systems, resulting in less surface electron traps. As electron trapping occurs on a nanosecond and slower timescale, these become in particular important when coupling NR to molecular catalysts such as hydrogenases, where electron transfer to the catalytic center occurs on a tens of nanosecond timescale [79] and trapping of electrons can block the transfer to a catalytic reaction center [80].

Implications for photocatalysis. These findings illustrate that the choice of the surface ligand should be considered with care in the design of assemblies for light-driven catalysis. In short, surface ligands impact the charge localization processes in NR drastically: thiolates induce hole traps, impacting sub-10-picosecond processes, and competing with the rod-to-seed localization, while bulky, sterically demanding ligands induce electron trapping sites due to undercoordinated Cd atoms, impacting the exciton recombination kinetics on a nanosecond timescale, possibly competing with electron transfer processes in this time range. Additionally, but not regarded in this report, not only surface coverage, but also length of the hydrophilic chain is known to negatively impact the electron transfer rate to a surface-proximal molecular catalyst [39,40].

Thus, NR with none or only a few surface traps (both hole and electron traps) seem to be desirable. However, this might not be universally true. While in the study at hand, DHLA-based systems showed the highest hole-trap efficiency and considerable electron traps, DHLA-capped CdSe QDs have already shown excellent photon-to-hydrogen yields in the order of several tens of percent [3,81,82]. Hence, under certain circumstances, surface trapping of charges can also support the desired function and increase their availability to surface-proximal catalytic centers and/or hole scavengers. A similar concept has recently been introduced as trap-state mediated charge transfer [83,84]. Thus, deciding on an optimal surface ligand is a sophisticated task and affords detailed knowledge on the function determining mechanistic steps and the impact of the ligand on these.

## 4. Materials and Methods

Chemicals and reagents. All chemicals and solvents were purchased from Sigma Aldrich (Merck KGaA, Darmstadt, Germany) except octadecylphosphonic adid (ODPA) were purchased from Carl Roth GmbH + Co. KG (Karlsruhe, Germany) and used without any further purification. All solvents were of spectroscopic grade.

Synthesis of CdSe seeds. Synthesis of the CdSe seed was adapted form a literature-known procedure [14]. A 25-mL-three-neck-flask was filled with 3.00 g trioctylphosphine oxide (TOPO), 0.28 g octadecylphosphonic adid (ODPA), 0.06 g CdO. The synthesis was conducted under constant stirring and an inert atmosphere if not indicated otherwise. The flask was heated up to 80 °C to melt the chemicals and evacuated to get rid of the water content in the chemicals. Once no more gas emerged from the solution, the flask was heated to 120 °C and kept evacuation for 30 min. After that, the flask was purged with $N_2$. The flask was then heated to 320 °C, upon which the solution turns colorless due to the complexation of Cd-ODPA, and then cooled down to 120 °C. Then a vacuum was applied until gas formation in the reaction mixture stopped to get rid of the water which is a side product of the complexation of Cd-ODPA (at least 2 h). Then, the flask was purged with $N_2$ again and heated up to 340 °C. Next, 0.058 g Se dissolved in 0.36 g trioctyl phosphine (TOP) were injected. After the injection, the heating was immediately stopped, and the flask was cooled down by $N_2$ airflow to accelerate the cooling speed. After the temperature of the mixture was cooled down to 90 °C, 5 mL of toluene was injected into the mixture. The seeds were cleaned by centrifugation with 10 mL toluene in 10 mL methanol for five times and then dissolved in toluene. The diameter of the CdSe quantum dots was determined from the energetic position of the lowest energy absorption peak [85] to be 2.2 nm.

Synthesis of TOPO-NR. TOPO-NR were synthesized following protocols reported in literature [13]. A 25-mL-three-neck-flask was filled with 3.35 g trioctylphosphine oxide (TOPO), 1.08 g octadecylphosphonic acid (ODPA), 0.207 g CdO, and 0.06 g n-propylphosphonic acid (PPA). The synthesis was conducted under constant stirring and an inert atmosphere if not noted otherwise. The flask was heated up to 80 °C until the reaction mixture melted and evacuated to remove residual water from the mixture. Once gas formation stopped, the flask was heated to 120 °C and kept evacuated for 30 min. After that, the flask was flooded with $N_2$. The flask was then heated to 320 °C, upon which the solution turns colorless due to the complexation of Cd-ODPA, and then cooled down to 120 °C. Next, a vacuum was applied until gas formation in the reaction mixture stopped (at least 2 h) to remove water, which is a side product of Cd-ODPA complexation. Then, the flask was flooded with $N_2$ again and heated up to 340 °C. Next, 1.5 g TOP and 0.05 g sulfur dissolved in 0.60 g TOP were injected. After 20 s, 2 mg of CdSe seeds (diameter = 2.2 nm) dissolved in 0.5 g TOP were injected. The reaction was allowed to stir for 10 min until the color of the solution turned from red to orange. 5 mL of toluene was injected into the mixture once the temperature dropped below the flashing point of toluene to stop solidification of the mixture. After precipitated by 10 mL methanol, rods were cleaned by centrifugation with 6 mL n-hexane, 2 mL nonanoic acid, and 2 mL octylamine in 10 mL methanol five times. Next, the size of the rods was excluded by centrifugation at 4200 rpm for 30 min with 10 mL toluene and 8 mL isopropanol to obtain rods with lengths of circa 30 nm. This was repeated two times in total. The NR were then dispersed in toluene for further investigation.

We synthesized two batches of TOPO-NR with a length of 30.2 ± 2.3 nm and 29.3 ± 2.7 nm, respectively, as determined by transmission electron microscopy (TEM) and a diameter of 4.6 ± 0.5 nm and 4.3 ± 0.5 nm, respectively. (Figure S1). The size distribution of the NR (length, width) from TEM images were determined using ImageJ 1.52a [86].

Ligand exchange with 11-mercaptoundecanoic acid (MUA). Ligand exchange with MUA was performed following a protocol described by Amirav and Alivisatos [4]. 250 mg MUA were dissolved in 25 mL methanol and tetramethylammonium hydroxide pentahydrate was added until the solution reached pH 11 (circa 200 mg). 20 mg TOPO-NR (dried from its toluene solution under vacuum) were added into this mixture and stirred for 2 h. When the NR were fully dispersed, approximately 35 mL of toluene was added as non-solvent until NR precipitated. The mixture was then centrifuged at 6000 rpm for 20 min, the supernatant discarded, and the precipitate was redispersed in degassed water.

Ligand exchange with poly(ethylene glycol) 2-mercaptoethyl methyl ether (HS-PEG-OCH$_3$). The ligand exchange procedure for HS-PEG-OCH$_3$ was similar to MUA (vide supra). 50 mg HS-PEG-OCH$_3$ ($M_n \approx 800$ g mol$^{-1}$, equal to roughly 18 repeating units) were dissolved in 5 mL methanol and tetramethylammonium hydroxide pentahydrate was added until the solution reached pH 11 (circa 200 mg). 4 mg TOPO-NR were added into this mixture and stirred for 2 h at room temperature. Different from the ligand exchange protocol for MUA, the mixture was then transferred into a centrifugation tube with a filter (Vivaspin 20 centrifugal concentrator with a membrane made of polyethersulfone with 10,000 molecular weight cut off) and then centrifuged at 6000 rpm for 20 min. The precipitate was redispersed in 5 mL methanol and centrifuged again under the same conditions. This was repeated three times in total. The precipitate was finally dispersed in degassed water.

Ligand exchange with dihydrolipoic acid (DHLA). The ligand exchange procedure for DHLA was similar to MUA (vide supra). 50 mg DHLA were dissolved in 5 mL methanol and tetramethylammonium hydroxide pentahydrate was added until the solution reached pH 11 (circa 200 mg). 4 mg dry TOPO-NR were added into this mixture and stirred for 2 h at room temperature. The mixture was then transferred into a centrifugation tube with a filter (Vivaspin 20 centrifugal concentrator with a membrane made of polyethersulfone with 10,000 molecular weight cut off) and then centrifuged at 6000 rpm for 20 min. The precipitate was redispersed in 5 mL methanol and centrifuged again under the same conditions. This was repeated three times in total. The precipitate was finally dispersed in degassed water.

Synthesis of dihydrolipoic acid–poly(ethylene glycol) ester (DHLA-PEG). Synthesis of DHLA-PEG followed a literature-known protocol by Uyeda et al. [41]. In short, poly(ethylene glycol) (PEG) (40g, $M_n \approx 400$ g mol$^{-1}$, equal to roughly 9 repeating unit) was attached onto lipoic acid (LA) (2.06 g, using a dicyclohexylcarbodiimide (DCC) (2.27 g) mediated esterification reaction with 4-(dimethylamino)-pyridine (DMAP) (0.37 g) as catalyst in dichloromethane (DCM, 100 ml) to obtain lipoic acid–poly(ethylene glycol) ester (LA-PEG). The precipitate that formed was filtered over a plug of Celite (SERVA Electrophoresis, Heidelberg, Germany) and the residual organic mixture washed with brine (75 mL) for three times to remove excess PEG. The combined organic extracts were dried over MgSO$_4$, filtered, and evaporated. The crude product was dissolved in DCM, purified by chromatography (ethyl acetate/methanol 95:5) and DCM was evaporated to obtain LA-PEG as a yellow oil. The reduction reaction was performed by adding NaBH$_4$ (0.083 g) into LA-PEG (1.21 g) in a 1:4 ethanol/water solution. To get pure DHLA-PEG, column chromatography (ethyl acetate/methanol 90:10) was applied and the residual solvent was evaporated.

Ligand exchange with dihydrolipoic acid–poly(ethylene glycol) ester (DHLA-PEG). For the ligand exchange, 50 mg DHLA-PEG was dissolved in 5 mL methanol and 4 mg dry TOPO-NR were added. The mixture was heated at 50 °C for 4 h under constant stirring. As DHLA-PEG-NR were dispersible in both polar and nonpolar solvents, the sample was purified three times using a centrifugal at 6000 rpm for 15 min. The precipitate was redispersed in degassed water.

Ligand exchange with hyperbranched poly(ethylene imine) (PEI). The ligand exchange with PEI followed the protocol developed by Thomas Nann for TOPO-capped CdSe@ZnS quantum dots [42]. 4 mg dry TOPO-NR and 50 mg hyperbranched PEI ($M_n = 25,000$ g mol$^{-1}$) were dispersed in 5 mL

$CHCl_3$. The mixture was stirred for 4 h followed by adding 10 mL cyclohexane to precipitate the NR. The precipitate was collected by centrifugation at 6000 rpm for 15 min and redispersed in degassed water.

Steady-state absorption spectroscopy. Absorption spectra were recorded in a quartz cell (d = 1 cm) using a JASCO V780 UV-Visible/NIR spectrophotometer (JASCO GmbH, Pfungstadt, Germany). All measurements were performed in a wavelength range of 200 nm to 800 nm. A cuvette with pure solvent was always measured as a reference.

Absorption coefficients $\varepsilon$ ere estimated on the basis of the absorption spectrum using a literature known procedure [47]. In short, the width of the *CdS* rods was estimated from the energy axis intercept obtained from a linear fit between 2.60 and 2.64 eV, yielding widths in the order of 4.9 nm, in good agreement with the TEM measurement. The size of the *CdSe* seed was determined from the peak position of *CdSe* absorption. Then, the *CdS* volume of a single NR $V_{CdS}$ was estimated by

$$V_{CdS} = \frac{1.337 \cdot A_{max,CdS} \cdot V_{CdSe}}{A_{max,CdSe}} \tag{2}$$

with the seed volume $V_{CdSe}$, the absorption at the peak position of *CdS* absorption $A_{max,CdS}$ and at the peak position of *CdSe* absorption $A_{max,CdSe}$. From the rod volume and width, a rod length of circa 35 nm was estimated, again in good agreement with the TEM results. Finally, the absorption coefficient at 3.54 eV (350 nm) was obtained by

$$\varepsilon_{3.54eV} = 28326.9 \cdot \frac{L}{mol \cdot cm \cdot nm^3} \cdot V_{total} \tag{3}$$

with the total rod volume $V_{total}$.

IR spectroscopy. IR spectra were recorded using a Bruker Tensor 27 system (Bruker Corporation, Billerica, MA, USA) equipped with a mid-IR source (4000 to 600 $cm^{-1}$) with attenuated total reflection (ATR) mode. The sample is was drop casted on a $CaF_2$ substrate (1 cm × 1 cm) and the respective solvent was evaporated under vacuum. The background of the $CaF_2$ substrate was always scanned before the measurements of the samples.

Steady-state photoluminescence spectroscopy. Photoluminescence spectra were recorded in a quartz cell (d = 1 cm) using an FLS980 photoluminescence spectrometer (Edinburgh Instruments Ltd., Livingston, UK) in a 90° geometry. An excitation wavelength of 400 nm was used to record photoluminescence spectra covering a wavelength range of 420 nm to 700 nm. The optical density of the dispersions was set to 0.05 to avoid inner filter effects and reabsorption of photoluminescence.

Photoluminescence spectra as a function of energy were corrected by scaling the measured intensity as a function of wavelength by the Jacobian transformation. For a quantitative description of the photoluminescence shape by means of fitting multiple Gaussians, the photoluminescence intensity was further divided by $E^3$ to obtain lineshape spectra with only little effect on the shape of the photoluminescence spectra [48]. Absolute photoluminescence quantum yields were recorded at an excitation wavelength of 450 nm and 500 nm using a barium sulfate coated integrating sphere [54,87]. The integrating sphere was mounted on the fluorimeter with the entry and output ports of the sphere located in 90° geometry from each other in the plane of the spectrometer. Photoluminescence was recorded from 425 to 700 nm. The colloidal samples were held in a five-face transparent quartz cuvette located in the center of the integrating sphere and the optical density at the excitation wavelength was adjusted to 0.1. As a reference sample, a cuvette filled with pure solvent was recorded under identical conditions.

Fs-transient-absorption spectroscopy. The fs time-resolved measurements were performed on as system consisting of a Ti: sapphire amplifier (Legend-Elite, Coherent Inc., Santa Clara, CA, USA), producing pulses centered at 795 nm with a repetition rate of 1 kHz and a pulse duration of <100 fs. The pump pulses centered at 390 nm were generated by second harmonic generation from the fundamental. A white light continuum with a spectral range from 340 to 750 nm was generated by focusing a

fraction of the fundamental into an eccentrically rotating $CaF_2$ crystal to probe the sample. The pump pulses were delayed with respect to the probe pulses by means of an optical delay stage (maximum delay: 2 ns) and focused into the sample by a lens (f = 75 cm). The repetition rate of the pump pulses was reduced to 500 Hz by a mechanical chopper and the polarization of the pump with respect to the probe pulses was set to the magic angle (54.7°) using a Berek compensator (Thorlabs GmbH, Bergkirchen, Germany) and a polarizer. The power density of the pump pulse at the sample position was adjusted to 0.1 W·cm$^{-2}$ and below, ensuring circa 2 photon absorption per NR or less. The white light continuum was split into probe and reference. The probe pulse was focused onto the sample by a concave mirror (f = 500 mm) and spatially overlapped with the pump pulse. Probe and reference were collected by a detection system (Pascher Instruments AB, Lund, Sweden) consisting of a spectrograph (Acton, Princeton Instruments, Trenton, NJ, USA) equipped with a double-stripe diode array detector. The diode array is read out with the laser repetition rate and the signal (ΔA) is calculated from two consecutive probe pulses, corresponding to pump-on and pump-off conditions. Samples were prepared and measured under inert conditions using home-built sealed, airtight cuvettes with 1 mm path length. The optical density of the sample was set to 0.4–0.6 at the excitation wavelength. Global fits were performed with a program provided by Jens Uhlig.

Time-resolved photoluminescence spectroscopy. Spectrally resolved photoluminescence decay curves were determined employing a Hamamatsu HPDTA streak camera (C4334, Hamamatsu Photonics K.K., Hamamatsu, Germany). Samples were excited by pulses centered at 390 nm created by frequency-doubling the output of a Ti: sapphire laser (Tsunami, Newport Spectra-Physics GmbH, Berlin, Germany). The repetition rate of the fundamental is reduced to 400 kHz by a pulse selector (model 3980, Newport Spectra-Physics GmbH, Berlin, Germany). Photoluminescence was collected for solutions from a 1 cm cuvette in a 90° angle and spectrally dispersed on the detector using a CHROMEX spectrograph (Chromex, Albuquerque, NM, USA). The optical density of the solutions was adjusted to circa 0.1 at the excitation wavelength. Measurements were performed with a polarizer set to magic angle, i.e., set to 54.7° with respect to the excitation polarization, in the detection path. A glass plate was used to record the instrumental response function (IRF) by reflecting parts of the attenuated excitation beam directly into the detector. IRFs were typically in the order of 0.3 ns. The emission was spectrally integrated and the resulting decay traces fitted with DecayFit software [88].

## 5. Conclusions

By combining results from steady-state and time-resolved spectroscopic techniques the influence of the type of surface ligand with respect to the anchoring group and steric demands on fundamental exciton relaxation and charge-carrier trapping processes in CdSe@CdS dot-in-rod nanostructures was investigated. The results of this investigation will support the design of optimized assemblies for light-driven catalysis incorporation CdSe@CdS dot-in-rod nanostructures as sensitizers. Via modulating the density and type of trapping states available the surface functionalization impacts, e.g., hole-localization driven rod-to-seed exciton localization, which occurs on a sub-10 ps time scale and has been shown to be precondition for the formation of long-lived charge separation in systems functionalized with metal nanoparticles acting as catalytic reaction centers and high photon-to-hydrogen conversion efficiencies. In such scenario PEI or ligands binding to the surface via amino groups in general seem to be a good choice, because they lead to efficient passivation of hole trapping sites and, as is shown in this study, high localization efficiencies of the exciton in the seed. On the other hand, trap states have been shown to be able to mediate charge transfer processes to molecular acceptors. In this case an increase of the density of trapping sites could support the targeted function. We find that steric restrictions, e.g., in dithiolates or ligands with bulky side chains (PEG), which decrease the surface coverage lead to an increase in the density of electron trap states, influencing the recombination dynamics at the ns timescale.

**Supplementary Materials:** The following are available online at http://www.mdpi.com/2073-4344/10/10/1143/s1, Figure S1. (A) and (D) TEM images of the two batches of NR used in this work together with (B) and (E)

length and (C) and (F) width distribution of the rods. Figure S2. IR-spectra of TOPO and TOPO-NR. Table S1. IR-frequencies of TOPO and TOPO-NR. Figure S3. IR-spectra of MUA and MUA-NR. Table S2. IR-frequencies of MUA and MUA-NR. Figure S4. IR-spectra of HS-PEG and HS-PEG-NR. Table S3. IR-frequencies on HS-PEG and HS-PEG-NR. Figure S5. IR-spectra of DHLA and DHLA-NR. Table S4. IR-frequencies of DHLA and DHLA-NR. Figure S6. IR-spectra of DHLA-PEG and DHLA-PEG-NR. Table S5. IR-frequencies of DHLA-PEG and DHLA-PEG-NR. Figure S7. IR-spectra of PEI and PEI-NR. Table S6. IR-frequencies on PEI and PEI-NR. Figure S8. Photoluminescence spectra of NR capped with different surface ligands normalized to their integrated area. Figure S9. Gaussian fits of steady-state spectra of TOPO-NR. Figure S10. Gaussian fits of steady-state spectra of MUA-NR. Figure S11. Gaussian fits of steady-state spectra of HS-PEG-NR. Figure S12. Gaussian fits of steady-state spectra of DHLA-NR. Figure S13. Gaussian fits of steady-state spectra of DHLA-PEG-NR. Figure S14. Gaussian fits of steady-state spectra of PEI-NR. Table S7. Fitting results for absorption spectra of nanorods with different surface ligands by five Gaussians. Table S8. Fitting results for line-shape spectra of nanorod photoluminescence with different surface ligands by two Gaussians. Figure S15. Gaussian fits of steady-state spectra of QDs used as seeds. Figure S16. Photoluminescence spectra of DHLA-NR synthesized under neutral and basic conditions recorded in water. Table S9. Localization and intrinsic radiative recombination efficiencies of the samples investigated, calculated via Equation (1). Figure S17. Transient absorption spectroscopy of TOPO-NR upon 390 nm excitation. Figure S18. Transient absorption spectroscopy of MUA-NR upon 390 nm excitation. Figure S19. Transient absorption spectroscopy of HS-PEG-NR upon 390 nm excitation. Figure S20. Transient absorption spectroscopy of DHLA-NR upon 390 nm excitation. Figure S21. Transient absorption spectroscopy of DHLA-PEG-NR upon 390 nm excitation. Figure S22. Transient absorption spectroscopy of PEI-NR upon 390 nm excitation. Figure S23. Spectroscopic evidence of hole trapping in the spectral region of 600–750 nm recorded for TOPO-NR in toluene. Figure S24. Relative contribution of DAS($\tau$n) to the total CdS bleach decay.

**Author Contributions:** Conceptualization, M.M. and M.W.; Methodology, M.M. and M.W.; Validation, M.M., B.L., and M.W.; Formal Analysis, M.M. and B.L.; Investigation, M.M. and B.L.; Resources, M.W.; Writing—Original Draft Preparation, M.M. and B.L.; Writing—Review & Editing, M.M. and M.W.; Visualization, M.M.; Supervision, M.W.; Project Administration, M.W.; Funding Acquisition, M.W. All authors have read and agreed to the published version of the manuscript.

**Funding:** Financial support is acknowledged by China Scholarship Council (CSC), the German Research Foundation (DFG)—project number 364549901—TRR234 (CataLight, B4 and Z2) and the Fonds der Chemischen Industrie (FCI).

**Acknowledgments:** M.M. gratefully acknowledges fruitful discussions with A. Schleusener and R. Baruah. We acknowledge the Leibniz Institute of Age Research–Fritz Lipmann Institute (FLI) Jena providing access to TEM and Katrin Buder (FLI) for help with TEM. The publication of this article was funded by the Open Access Fund of the Leibniz Association.

**Conflicts of Interest:** The authors declare no conflict of interest.

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
