# Peer review of "Influence of Surface Ligands on Charge-Carrier Trapping and Relaxation in Water-Soluble CdSe@CdS Nanorods"

_catalysts, doi:10.3390/catal10101143_

Round 1
Reviewer 1 Report
In this manuscript, the authors present their comprehensive study on the charge carrier trapping and relaxation process in water-soluble CdSe@CdS dot-in-rod nanorods with commonly employed ligands. The steady-state and time-resolved photoluminescence and absorption spectra demonstrate the charge carrier trapping and relaxation processes in CdSe@CdS nanorods with different ligands. I think that the results are interesting and the findings are useful for light-driven hydrogen generation. Overall, the paper is well organized and well written. I would like to recommend the publication of this work in Catalysts.
Author Response
We thank the reviewer for the very positive evaluation.
Reviewer 2 Report
Comments:
In the article “Influence of Surface Ligands on Charge-Carrier Trapping and Relaxation in Water-Soluble CdSe@CdS Nanorods” authors report a detailed investigation of steady-state and time-resolved photoinduced excitation dynamics of water-soluble nanorods with various anchor groups and the results were compared with the standard benchmark system of the same nanorods with TOPO. The authors have deeply characterized the material with different photophysical techniques which are interesting, however, considering the aim and scope of the Journal, the authors only hypothesized the materials and the role of capping agents for its applicability in catalysis, and they did not carry out any catalysis experiment with the material nor show the effect of different capping agents in the catalysis to back their claim. Hence, the work doesn’t perfectly fit the journal scope.
Minor comments
- Abstract line 24= phosphine oxid to be changed to “phosphine oxide”
- On page no.1 lines 37-39; the author has to include the reference.
- On page no. 2 lines 53; the author mentioned “(in this work, LIGAND-NR will be” why ligand mentioned here in capital letters, is there any specific reason for this.
- The author report dimension of the NR without error on page no.3, the author has to report the values with error wherever it is possible.
- On page no.4; the author mentioned “the absorption spectrum of all NR is characterized by different lowest exciton transition between valence and conduction band levels ….” to claim such a statement author has to provide references for that statement.
- S1 and Fig. S8 nowhere mentioned in the main text.
- Table 1. Caption have to be reduced, there is a lot of redundancy.
- Experimental section, line 553 = 0.058 g CdSe, What is the selenium source in this synthesis?
- Figure 2b = all ligand NR, PL spectra seems to be blueshifted to TOPO-NR, However, contrary to this, Table 1 shows DHLA-PEG- NR red-shifted by 20 meV. Similarly, Line 182, the hypsochromic shift, and the value (2.14 eV) are different from the table. The authors need to explain or correct it.
- Line 260,261, The authors say redshift, however, it is only a blue shift observed in Fig 2b.
- Authors need to explain why there is a shift after the ligand exchange, For example, How does PEI-NR show a high blue shift and good QY, compared to others?.
- The conclusion section should change as a summary and a conclusion or discussion.
Based on such arguments, I recommend minor revision before its publication in Catalysts at this current form of the manuscript.
Author Response
We thank the reviewer for his careful reading and valuable comments. We followed the reviewer’s suggestions and corrected the manuscript accordingly. Replies to the questions and the changes made are documented in detail below.
The reviewer states:
“...the authors only hypothesized the materials and the role of capping agents for its applicability in catalysis, and they did not carry out any catalysis experiment with the material nor show the effect of different capping agents in the catalysis to back their claim.”
The relevance of these structures as sensitizers in light-driven hydrogen generation is well documented in literature and references are cited in the introduction of the manuscript accordingly. Also it has been shown that the surface ligand plays role in tuning the activities of the systems. Although known, the source of these effects has not been studied in detail yet. To gain insight, we focused in our study on the sensitizer itself and how surface functionalization is changing the basic light-induced processes, i.e. exciton localization and charge-carrier dissociation and trapping processes in these structures. These processes are the fundament for the application of these structures in light-driven catalysis. Hence we believe, although no catalysis shown itself in this study, the topic matches the journal and especially the special issue on “Photo-Induced Electron Transfer Kinetics in Catalysis”.
Corrections requested:
- Abstract line 24= phosphine oxid to be changed to “phosphine oxide”
Response: The typographical error has been corrected.
- On page no.1 lines 37-39; the author has to include the reference.
Response: We apologize for the missing refecences. The following references were added to substantiate the statement of the usability of CdSe@CdS nanorods as photosensitizers in the HER (p. 1, l. 39):
- Kalisman, P.; Houben, L.; Aronovitch, E.; Kauffmann, Y.; Bar-Sadan, M.; Amirav, L. The golden gate to photocatalytic hydrogen production. J. Mater. Chem. A 2015, 3, 19679–19682, doi:10.1039/C5TA05784A.
- Wu, K.; Zhu, H.; Lian, T. Ultrafast Exciton Dynamics and Light-Driven H2 Evolution in Colloidal Semiconductor Nanorods and Pt-Tipped Nanorods. Acc. Chem. Res. 2015, 48, 851–859, doi:10.1021/ar500398g.
- Qiu, F.; Han, Z.; Peterson, J.J.; Odoi, M.Y.; Sowers, K.L.; Krauss, T.D. Photocatalytic Hydrogen Generation by CdSe/CdS Nanoparticles. Nano Lett. 2016, 16, 5347–5352, doi:10.1021/acs.nanolett.6b01087.
- Amirav, L.; Alivisatos, A.P. Photocatalytic Hydrogen Production with Tunable Nanorod Heterostructures. J. Phys. Chem. Lett. 2010, 1, 1051–1054, doi:10.1021/jz100075c.
- On page no. 2 lines 53; the author mentioned “(in this work, LIGAND-NR will be” why ligand mentioned here in capital letters, is there any specific reason for this.”
Response: In the sentence on page 2 line 53/54 we introduce the nomenclature used throughout the manuscript distinguishing between free ligand, where just the abbreviation for the ligand name (symbolized by LIGAND) will be used, and ligand capped rods, which we abbreviate with LIGAND-NR. LIGAND in capital letters stands here for a generic ligand name, to be substituted by TOPO, MUA, DHLA, HS-PEG and DHLA-PEG respectively. We felt the distinction to be important, as both intrinsic ligand properties as well as ligand-capped NR properties are discussed throughout the paper.
- The author report dimension of the NR without error on page no.3, the author has to report the values with error wherever it is possible.”
Response: The exact NR dimensions were reported in the Materials and Methods section, but are now also included in the main text (p. 3, l. 129). We inserted:
“...yielding TOPO-NR of a length of c. 30 nm and a diameter of c. 4.5 nm (for the two batches investigated: 30.2 ± 2.3 nm and 4.6 ± 0.5 nm, and 29.3 ± 2.7 nm 4.3 ± 0.5 nm; Figure S1).
- On page no.4; the author mentioned “the absorption spectrum of all NR is characterized by different lowest exciton transition between valence and conduction band levels ….” to claim such a statement author has to provide references for that statement.”
Response: A reference to support this statement was added on p. 4 l. 153:
- Wu, K.; Rodríguez-Córdoba, W.E.; Liu, Z.; Zhu, H.; Lian, T. Beyond Band Alignment: Hole Localization Driven Formation of Three Spatially Separated Long-Lived Exciton States in CdSe/CdS Nanorods. ACS Nano 2013, 7, 7173–7185, doi:10.1021/nn402597p.
The detailed assignment to these transitions based on our analysis follows in the subsequent paragraph.
- S1 and Fig. S8 nowhere mentioned in the main text.
Response: Fig. S1 was previously only mentioned in the Materials and Methods section but is now also referenced in the main text (p. 3, l. 129). Figure S8 was already referenced in the main text in the caption of Figure 2. As it only serves as a different way of data visualization, it is not explicitly discussed in the main text.
- Table 1. Caption have to be reduced, there is a lot of redundancy.
Response: The referee is right to point out that the caption of Table 1 is verbose, maybe to the point of redundancy if taken together with the main text and the Materials and Methods section, yet we believe that the information in the caption helps the reader to properly understand the table without cross-checking these sections and to avoid confusion. We reduced the caption to:
“Steady-state absorption and photoluminescence data of NR with different surface ligands in pure water, TOPO-NR in toluene: Eabs,CdS and Eabs,CdSe spectral position of the lowest energy excitonic CdS and CdSe seed absorption peak position, EPL spectral position of the peak photoluminescence upon excitation at 3.10 eV (400 nm), and the absolute photoluminescence quantum yields ΦPL,450 and ΦPL,500 upon excitation at 2.76 eV (450 nm) and 2.48 eV (500 nm), respectively. For ΦPL,500, the error is equal to the instrumental error.”
- Experimental section, line 553 = 0.058 g CdSe, What is the selenium source in this synthesis?”
Response: We thank the reviewer for discovering this mistake. Elemental Se dissolved in TOP serves as the selenium source. The line now reads:
“Next, 0.058 g Se dissolved in 0.36 g TOP were injected.”
- Figure 2b = all ligand NR, PL spectra seems to be blueshifted to TOPO-NR, However, contrary to this, Table 1 shows DHLA-PEG- NR red-shifted by 20 meV. Similarly, Line 182, the hypsochromic shift, and the value (2.14 eV) are different from the table. The authors need to explain or correct it.”
Response: We thank the referee for noticing this error. The referee is correct that PL spectra of MUA-NR, DHLA-NR, and HS-PEG-NR are slightly blue-shifted compared to TOPO-NR. This has been corrected in the text. Further, the spectrum of DHLA-PEG-NR in Figure 2b was plotted against wrong energy values. This has been corrected in the figure. Now, the red-shift of the emission is clearly visible. This has also been corrected in Figure S8. We apologize for this mistake. The shift of PL for PEI-NR is now correctly given in the main text as 30 meV to a position of 2.19 eV. We also corrected an erroneous “hypsochromical” to “hypsochromic”.
The corrected paragraph reads now (p. 5 l. 181 ff.):
“For MUA-NR, HS-PEG-NR, and DHLA-NR, the spectral shape remains largely unaffected, while the peak position of photoluminescence blueshifts by 10-20 meV. For PEI-coating, the hypsochromic shift of the photoluminescence peak is even more pronounced (30 meV to 2.19 eV), while DHLA-PEG-NR exhibit a bathochromic shift of the photoluminescence peak by c. 20 meV.”
- Line 260,261: The authors say redshift, however, it is only a blue shift observed in Fig 2b.”
Response: As stated in the response to comment 9, the depiction of DHLA-PEG-NR in Figure 2b was erroneous. We corrected this and now the red-shift of PL is reported correctly for DHLA-PEG-NR. We apologize for this mistake.
- Authors need to explain why there is a shift after the ligand exchange, For example, How does PEI-NR show a high blue shift and good QY, compared to others.
Response: The changed photoluminescence behavior of PEI-NR is explained in the discussion of the photoluminescence results in the main text (p. 7, l. 279 ff.):
“PEI-NR behave qualitatively different from the other investigated ligands, as they show not only an enhancement of ΦPL and Φloc to near unity, but also a hypsochromic shift of both absorption and photoluminescence by 20 meV. Notably, this shift also includes the exciton absorption related to the CdSe core. While previous studies on alkylamine-capped CdSe quantum dots assigned these changes to etching of Cd surface atoms, a recent report by Morgan and Kelley proposed that amines do not merely passivate surface atoms, but shifts the energetic levels of both valence and conduction band, making hole traps inaccessible [62]. This results in the blue-shift of absorption and emission and a near-unity photoluminescence.
The referee is correct in stating that for the other ligands, we did not make an explicit statement on reasons for the observed minor shift. The reason for the apparent shift of the PL spectra can be rationalized on the different weights of the two states we disentangled by the bimodal Gaussian fit of the PL spectra. For example, the high-energy PL state of DHLA-PEG-NR contributes less to the overall PL spectrum than the same state in, e.g., TOPO-NR, leading to a red-shift. We added a sentence commenting on this effect (p. 5, l. 210 ff.):
“The apparent shift of the photoluminescence maximum can then be explained by the different weights of the two photoluminescent states to the entire photoluminescence spectrum.”
- The conclusion section should change as a summary and a conclusion or discussion
Response: The section was changed to discussion. A short Conclusion section has been added at the end of the manuscript.
“5. Conclusion
By combining results from steady-state and time-resolved spectroscopic techniques the influence of the type of surface ligand with respect to the anchoring group and steric demands on fundamental exciton relaxation and charge-carrier trapping processes in CdSe@CdS dot-in-rod nanostructures was investigated. The results of this investigation will support the design of optimized assemblies for light-driven catalysis incorporation CdSe@CdS dot-in-rod nanostructures as sensitizers. Via modulating the density and type of trapping states available the surface functionalization impacts, e.g. hole-localization driven rod-to-seed exciton localization, which occurs on a sub-10 ps time scale and has been shown to be precondition for the formation of long-lived charge separation in systems functionalized with metal nanoparticles acting as catalytic reaction centers and high photon-to-hydrogen conversion efficiencies. In such scenario PEI or ligands binding to the surface via amino groups in general seem to be a good choice, because they lead to efficient passivation of hole trapping sites and, as is shown in this study, high localization efficiencies of the exciton in the seed. On the other hand, trap states have been shown to be able to mediate charge transfer processes to molecular acceptors. In this case an increase of the density of trapping sites could support the targeted function. We find that steric restrictions, e.g. in dithiolates or ligands with bulky side chains (PEG), which decrease the surface coverage lead to an increase in the density of electron trap states influencing the recombination dynamics at the ns timescale.”